

# Microbial communities inhabiting 600-year-old sediments in the Inka-Coya Lake located in the Atacama Desert

Coral Pardo-Esté[1]; Juan Castro-Severyn[2]; Francisco Remonsellez[2,3]; Antonio Maldonado[4,5]; Inger Heine Fuster[6]; Hector Pizarro[7]; Adriana Aránguiz-Acuña[4,6]*.

[1] Departamento de Ciencias Farmacéuticas, Facultad de Ciencias, Universidad Católica del Norte, Antofagasta, Chile.
[2] Laboratorio de Microbiología Aplicada y Extremófilos, Departamento de Ingeniería Química, Universidad Católica del Norte, Antofagasta, Chile.
[3] Centro de Investigación Tecnológica del Agua en el Desierto-CEITSAZA, Universidad Católica del Norte, Antofagasta, Chile.
[4] Millennium Nucleus of Andean Peatlands, AndesPeat.
[5] Centro de Estudios Avanzados en Zonas Áridas, CEAZA. La Serena, Chile.
[6] Laboratorio de Ecología Acuática, Departamento de Recursos Ambientales, Universidad de Tarapacá, Arica, Chile.
[7] Departamento de Ciencias Geológicas Universidad Católica del Norte, Antofagasta, Chile.

*Correspondence to*: Adriana Aránguiz-Acuña (aaranguiza@academicos.uta.cl)

**Abstract.** Lacustrine sediments are natural archives for the surrounding area's biogeochemical dynamics; in particular, the isolation and extreme conditions in which desert lakes are located make them ideal study models for studying perturbations in the ecosystem. Specifically, Inka-Coya Lake is in the Atacama Desert, where the presence of metals and metalloids associated with the active geological activity and local mining industry is a crucial driver for the biological dynamics in this ecosystem, as have been suggested for macroinvertebrates and plankton communities in the lake. In this study, we aimed to characterize the microbial communities that inhabit deep lacustrine sediments and their interaction with the surrounding environment. The results show that the microbial community from lacustrine sediments contains over 70% unclassified organisms, highlighting this ecosystem's microbial taxonomic novelty. Our results indicate that the microbial communities cluster in three distinct zones: a superficial community, an intermediate and mixed community, and a more specialized anaerobic community in the deeper sediments. The microbial composition is dominated by chemoheterotrophic bacteria strongly associated with methane metabolism. Additionally, there is statistical evidence of strong correlations between particular taxa such as Sulfurimonadaceae, Metanoregulaceae, and Ktedonobacteroceae with elements like Cu, As, Fe, Ni, and V, and magnetic properties of the surrounding environment. Further detailed studies of the metabolic repertoire of these communities are necessary to understand the complex dynamics between microbial life and geochemical composition in this fragile and extreme environment.

**KEYWORDS**: desert lake, deep lacustrine sediment, microbial communities, extremophiles



## 1. INTRODUCTION

The Atacama Desert is located on the western slopes of the Central Andes Cordillera between 15 and 30°S at elevations
between sea level and 3,500 m a.s.l., in the driest part of the South American dry diagonal, which extends from 5°S on the
west coast to almost 50°S on the east coast, over 4,000 km with less than 200 mm mean annual rainfall (MAR). The Andes
Cordillera represents a physical barrier that directly modulates climatic conditions and water availability in the Atacama Desert
(Garreaud et al., 2003). Particular environmental conditions of the Atacama, such as high solar radiation, low atmospheric
humidity, and other pressures associated with the natural composition of the desert, directly influence life occurring in these
ecosystems (Demergasso et al., 2008; Albarracín et al., 2020; Kurth et al., 2021; Kereszturi et al., 2020; Borsodi et al.,
2022). Thus, the Atacama Desert water bodies are truly natural laboratories for understanding evolutionary processes, not
only of the geomorphology of the landscape but also of different life forms promoted by environmental forces, such as climate
changes at different time scales (Adrian et al., 2009).
Aquatic sediments are sources and/or sinks of elements participating in biogeochemical cycles, including both allochthonous
and autochthonous lake processes, influencing biodiversity and trophic dynamics of water bodies (Trolle et al., 2010;
Fernández et al., 2000; Usenko et al., 2007; Bandowe et al., 2018). Aquatic sediments have an advantage over terrestrial
records, and that is that they accumulate at measurable rates because they are often buffered from physical, chemical, and
biotic disturbances, thus allowing the recording of past environmental conditions (Benito, 2020; Picard et al., 2022; Da Costa
et al., 2023; Yan et al., 2024). A wide variety of abiotic (e.g., bulk density, dry mass, radioactive isotopes, mineralogy, chemical
elements) and biotic proxies (e.g., fossils, species abundance, and presence/absence, resting structures, pigments,
environmental DNA) preserved in the sediments are currently analyzed to reconstruct ecosystem change at timescales ranging
from fine-scale (interannual or decadal) to millennial (Cohen, 2003; Korosi et al., 2017).
The central Andes water systems have mainly originated after successive glaciations and volcanic and tectonic activity. In the
Atacama Desert, lacustrine sediments are natural archives holding evidence of past precipitations, dust deposition, anthropic
disturbances, and pollution, mainly due to mining activity that occurs in the area (Grosjean and Veit, 2005; Placzek, 2009;
Cerda et al., 2019; Aránguiz-Acuña et al., 2020). Desert lakes are located in extremely arid and isolated areas, making them
susceptible to perturbations (Valero-Garcés et al., 2003; Pueyo et al., 2011; Grosjean and Veit, 2005) and hosting extreme
forms of life, especially microorganisms that have evolved physiological and life-history adaptations allowing them to perform
in challenging conditions (Dib et al., 2009; Ordoñez et al., 2009; Farías et al., 2013, 2014; Rasuk et al., 2014). Therefore,
lacustrine sediments are expected to contain a great taxonomic diversity, including low-abundance and highly specialized taxa,
directly influenced by small-scale conditions determining local environments (Borsodi et al., 2022).
Inka-Coya Lake (22°20'S-68°35'W, 2534 m.a.s.l.) is located at the eastern margin of the Atacama Desert, close to the Salado
River, in the San Francisco de Chiu-Chiu village, northern Chile. It is a karstic sinkhole developed during the Quaternary
period by the dissolution of calcareous layers of the Chiu-Chiu Formation (El Loa Group). The Atacama Desert and its
surroundings have a particular geological history; metals and metalloids found in the Inka-Coya area include Ti, Al, Fe, Ni,
and Cr; also, As and Sb are associated with the local geological activity (Aszalós et al., 2020; Borsdorf and Stadel, 2015;
Pérez-Portilla et al., 2024), that directly influence the chemical composition of the underground water (Vignale et al., 2021).
The central Atacama Desert, specifically the Antofagasta Region, holds large porphyry copper deposits that support the great
metal-mining industry (Dittmar, 2004; Salvarredy-Aranguren et al., 2008). In 2021, Chile was the world's top copper producer,
producing 5,508,084 tons -26.6% of the world's production- (Rodríguez-Luna et al., 2022). The mining industry in the
Antofagasta region has developed extensively since the 19th century (Dittmar, 2004; Salvarredy-Aranguren et al., 2008) with
increasing impacts on the national economy and development, but also on the health of ecosystems, which have triggered





social and environmental conflicts, affecting especially relevant groups such as ancient Indigenous communities from Quechua
and Lickan Antay people (see Ramírez et al., 2005; Tapia et al., 2019).
Previous studies aimed to assess mining pollution records on environmental matrices have included the sediment records of
different longitudes from the Inka-Coya Lake, showing variation through the geochemistry and magnetic properties (Cerda et
al., 2019; Aránguiz-Acuña et al., 2020; Pérez-Portilla et al., 2024). Analyzed variables have allowed us to identify episodes
associated with changes in water availability, flash flooding, and evidence of perturbations induced by mining activities.
Overall, the lake is polluted at different degrees of severity with Cu, Sb, Mo, and As, and some elements like Cu and Ni have
been enriched in the most recent periods (Cerda et al., 2019; Pérez-Portilla et al., 2024). Additionally, some biological proxies,
such as macroinvertebrates and diatom communities, were found to be directly influenced by the accumulation of metal(loid)
as observed by changes in assemble composition (Aránguiz-Acuña et al., 2020). Surrounding metal-mining exploitation,
which has been maintained and even increased through the last 200 years, in addition to aridity stable conditions, makes Inka-
Coya Lake an excellent site for understanding biological adaptations of aquatic populations to these anthropic pressures
(Aránguiz-Acuña et al., 2018; 2020).
While microbial life in arid ecosystems plays a key role in maintaining biogeochemical cycles (Madsen, 2011), there is a high
proportion of unclassified taxa that hold great interest in poly-extreme environments from an ecological, environmental, and
biotechnological point of view (Farias et al., 2014; Castro-Severyn et al., 2021; Dong et al., 2022). Nevertheless, there are
scarce records in which microbial assemblage has been used as a paleolimnological proxy of the possible responses to long-
term sustained anthropogenic metal stress (Da Costa et al., 2023; Yan et al., 2024). The few studies considering this aim have
focused on changes in primary producers' abundance, under-interpreting the impact on other metabolic functional groups
(Benito et al., 2020; Picard et al., 2022).
This study aimed to characterize the microbial community along a lacustrine sediment core obtained from Inka-Coya Lake.
To our knowledge, this is a pioneering study in the microbial characterization of a sedimentary core of this length (136 cm)
and date (600 years) from a lake sediment in the Atacama Desert. Results show that the microbial communities have changed
through time, identifying three clear periods in which alpha and beta diversity has been associated with organic matter content,
magnetic susceptibility, and metals and metalloid concentrations. Additional studies of the metabolic functions of the
microorganisms inhabiting these sediments are required to understand the interactions between microbial life and the
geochemical components of the Inka-Coya Lake further.

## 1. METHODS

### 2.1 Study site and sampling

Inka-Coya Lake (San Francisco de Chiu Chiu village, Antofagasta; 22° 20.300′ S; 068°35.981′ W, Chile) has a surface area of
500 m$^2$ and a maximum depth of 18 m, is located in the Pre-Andean Depression of the Antofagasta Region at an elevation of



2,520 m a.s.l. (Fig. 1 A, B). Around the lake, the predominant vegetation is of vegas, a type of wetland typical of the Andean
pre-Puna zone, strongly associated with the hydric variability of the emerging groundwaters.

## 2.2 Sediment Core Sampling

A fieldwork campaign was driven in August 2021. The topography of the lake bottom was modeled using the Echo-Map Plus
42CV from Garmin. Afterward, three sediment cores from the depocenter of Inka-Coya Lake, where maximum sedimentation
rates are expected, were obtained. The cores were obtained using a 9.0 cm diameter Uwitec gravity corer.  This study shows
the most extended core analysis results, measuring 136 cm (labeled LIC-SHC03). X-ray and photography images were
captured before obtaining sections of sediment subsamples from the core. For the geochemical analyses, sediment sub-samples
every 0.5 cm to a depth of 12 cm were obtained. Then, every 1 cm until the end of the core, totaling 146 sediment samples.
Additionally, sub-samples for every 1 cm interval were obtained to develop the magnetic susceptibility analysis. The cores'
detailed treatment, geochemical analysis, and magnetic properties can be reviewed by Pérez-Portilla et al. (2024).

## 2.3 Sediment Core Dating

The geochronology of the sediment core from Inka-Coya Lake was determined through radiocarbon dating (14C) on the
remaining macroscopic carbon along the record. The measurements were done using accelerator mass spectrometry (AMS),
and the results were corrected for isotopic fractionation with an unreported $\delta13C$ value. Subsequently, the age-depth model
for this sedimentary core was established using the Bayesian radiocarbon chronology package Bchron in R, using the 'shcal20'
as the calibration curve (Hogg et al., 2020; Haslett and Parnell, 2008). A detailed description of the procedure is available in
Pérez-Portilla et al. (2024).

## 2.4 Magnetic and Geochemical Properties Analysis

Five grams of each sediment sub-sample were placed into paleomagnetic boxes of 8 cm$^3$ to measure the mass magnetic
susceptibility ($\chi$) using a Kappabridge MFK1_FA instrument (AGICO Co) under environmental conditions (22–24°C) and a
magnetic field of 200 A/m. The samples were measured at a low frequency of 976 Hz ($\chi$lf or simply $\chi$) and a high frequency
of 15,616 Hz ($\chi$hf). The magnetic susceptibility dependent on the frequency was calculated using both measurements, as
described by Pérez-Portilla et al. (2024). The $\chi$fd% parameter is used to indicate the presence of magnetic particles near the
limit of the superparamagnetic/single domain (SP/SD) magnetic size (Verosub and Roberts, 1995), which can be linked to the
presence of magnetic particles of authigenic origin (Dearing et al., 1996). Additionally, sub-samples of each 1 cm slice were
dried in an oven at 50°C. Afterwards, they were homogenized using an agate mortar in the Geochemistry Laboratory of
Universidad Católica del Norte (UCN), Antofagasta, Chile. The sediments were then digested using reverse aqua regia (4 mL
HCl + 12 mL of HNO$_3$ + 300 mg of sediment sample) and a microwave digester (Perkin Elmer MPS 320; EPA 3052 method),
following Tapia et al. (2022) in Centro de Investigación Tecnológica del Agua en el Desierto (CEITSAZA-UCN). The
elements aluminum (Al), titanium (Ti), vanadium (V), manganese (Mn), iron (Fe), nickel (Ni), Cu, zinc (Zn), arsenic (As),
molybdenum (Mo), and antimony (Sb) were measured by the inductively coupled plasma atomic emission spectroscopy (ICP-
OES) Perkin Elmer Optima 7000 in the digested residue at CEITSAZA. The organic, inorganic matter, and carbonate contents
were estimated using the loss on ignition (LOI) method, which was assessed at a contiguous 1 cm interval following Heiri et
al. (2001). This procedure involved drying 1 cm$^3$ of each sediment sample in crucibles at 105°C for two hours and weighing
them. The dry samples were weighed before heating to 550°C in a flask over 1.5 h, left at 550°C for two h, then allowed to



cool. The samples were weighed, then the crucibles were transferred to the flask and burned at 925°C. Finally, the crucibles
were weighed again once they cooled.

## 2.5 Sediment sample processing and DNA extraction

Sediment samples were obtained every 1 cm from the top to the bottom of the core. According to the manufacturer's
instructions, total DNA was extracted from the 250 mg of sediment samples using the DNeasy PowerSoil kit (Qiagen Inc.,
Hilden, Germany). DNA integrity, quality, and quantity were verified by 1% agarose gel electrophoresis and fluorescence
using a Qubit 3.0 fluorometer and the Qubit dsDNA HS assay kit (Thermo Fisher Scientific, MA, USA). Following, DNA
samples were sent to AustralOmics, Chile, for amplification of the bacterial 16S rRNA gene V4 region (~ 250 bp) using the
515F and 806R primers (Caporaso et al., 2011), construction of 250 bp paired-end libraries and sequencing on a MiSeq
(Illumina) platform.

## 2.6 Taxonomic Composition Analysis

This analysis was conducted in R v4.0.3 and RStudio v1.3.1093 following the DADA2 v1.16.0 R package pipeline (Callahan
et al., 2016) to infer amplicon sequence variants (ASVs) for each sub-sample. Briefly, the reads were evaluated for quality
control and subsequently trimmed (Ns = 0, length ≥ 150 bp, expected errors ≤ 2), followed by dereplication, denoising, and
merging of paired reads. Following, an ASV table was built to allow a maximum of two expected errors, removing chimeras
and assigning taxonomy using the Silva v138 database (Quast et al., 2012). Also, all ASVs identified as Eukarya, Chloroplast,
and Mitochondria were removed. A multi-sequence alignment was created to infer phylogeny using FastTree v2.1.10 (Price et
al., 2009), and phyloseq-object (containing the ASVs, taxonomy assignment, phylogenetic tree, and the samples meta-data)
was created using the R package Phyloseq v1.34.0 (McMurdie et al., 2013). Finally, taxa relative abundance and taxonomic
composition at different ranks were visualized using the ggplot2 v3.3.3 (Wickham, 2016), Fantaxtic v0.2.0 (Teunisse, 2022),
and ampvis2 v2.7.4 (Andersen et al., 2018) R packages.

## 2.7 Diversity Analysis

Alpha diversity metrics (Shannon, Chao, phylogenetic diversity, and Simpson indexes) were calculated for each segment along
the core using the microbiome v1.24.0 (Lahti et al., 2017) and btools v0.0.1 R packages. Also, Wilcoxon statistical tests to
compare means between the identified zones were carried out and visualized using the DESeq2 v1.42.0 (Love et al., 2014) and
ggpubr v0.6.0 (Kassambara, 2017). Moreover, beta diversity was evaluated by principal coordinates analysis using Hellinger
transformed Bray Curtis distances based on the ASV abundance matrix were calculated using Phyloseq v1.34.0 (McMurdie et
al., 2013) and ampvis2 v2.4.5 (Andersen et al., 2018) R package. Also, redundancy analysis (RDAs) was calculated using



depth gradient and zone parameters to constrain the multivariate space, and ANOVA tested the statistical significance of the
selected geochemical variables.

## 2.8 Functional Predictions

Functional potential signatures and metabolic pathways abundances were predicted based on the ASV abundance and
taxonomy matrices using PICRUSt2 v2.4.1 software (Douglas et al., 2020) through the Kyoto Encyclopedia of Genes and
Genomes (KEGG) (Kanehisa et al., 2012) and MetaCyc (Caspi et al., 2018) pathway databases. The analysis of Differential
Abundance represented pathways was calculated using the Kruskal-Wallis test (confidence interval = 0.95) and the Benjamini-
Hochberg correction false-discovery rate using ggpicrust2 v1.7.2 R package (Yang et al., 2023). Also, we use the Functional
Annotation of Prokaryotic Taxa (FAPROTAX) database v1.2.7 (Louca et al., 2016) to map the identified ASVs and quantify
changes in established ecologically relevant functions.

## 1.   RESULTS

The samples analyzed correspond to a sediment core from Lake Inka-Coya, located in the Atacama Desert, with a water depth
of 18.5 meters (Fig. 1). The sediment core age-depth model was constructed based on six charcoal sample dates, where the
more superficial at 41 cm corresponds to $75 \pm 32$ cal years BP, and the deepest found at 94 cm corresponds to an age value of
$505 \pm 22$ cal years BP. Based on the age-depth model constructed, the sediment core of Inka-Coya Lake analyzed had 630
years of age (Fig. 2).



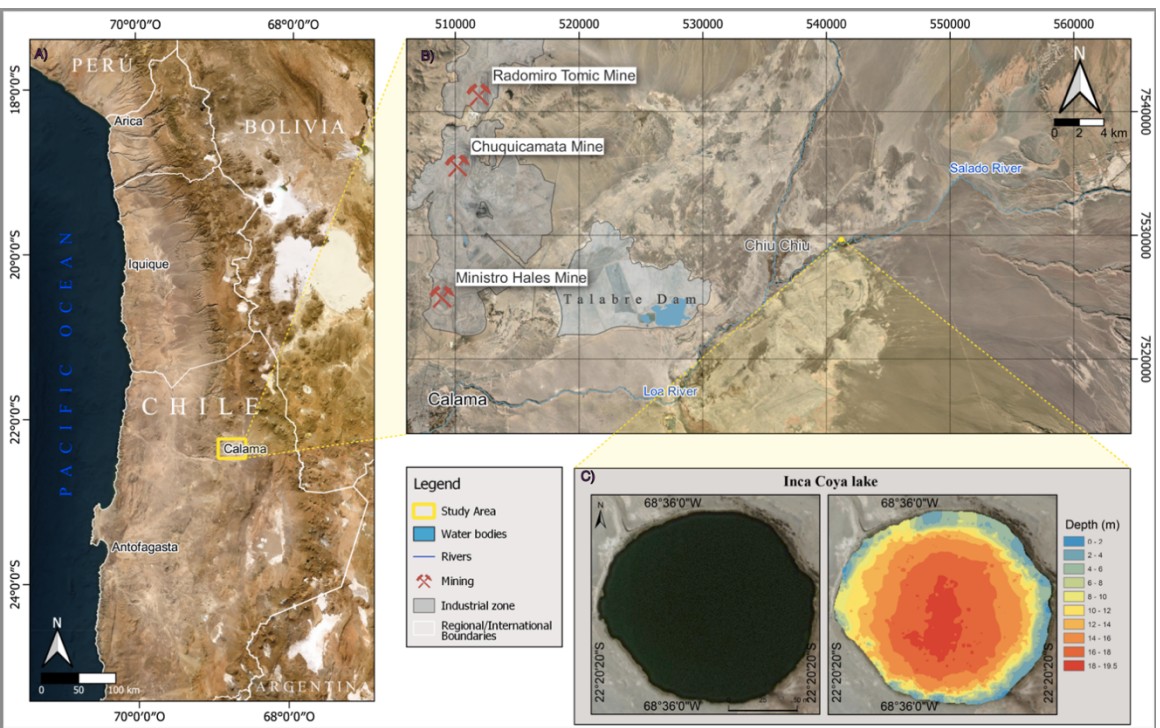

**Figure 1. Sampling site location in Northern Chile (A), Inka-Coya Lake, and important surrounding mining and urban centers (B) and the bathymetry of the lake (C).**





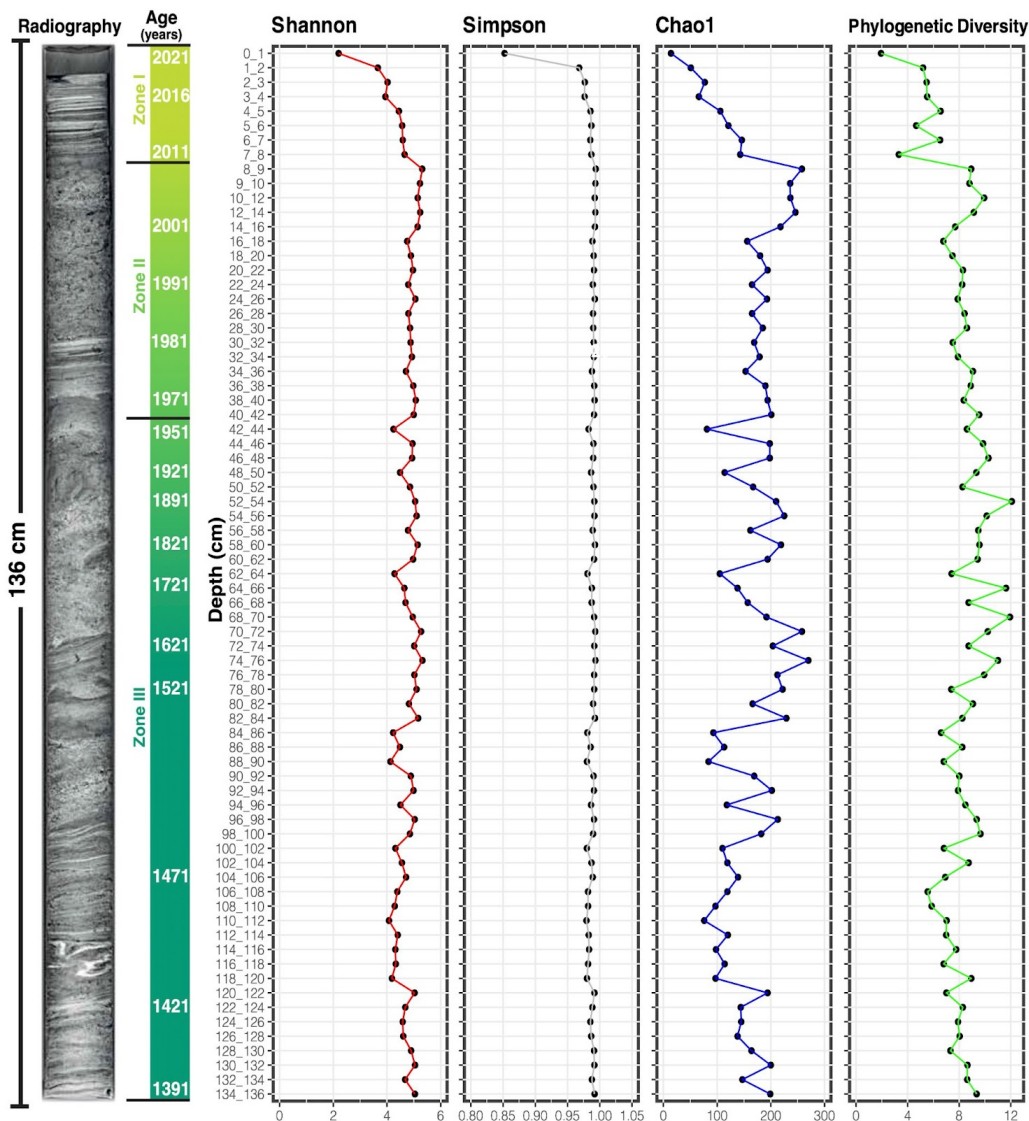

191

**Figure 2. Diversity of the microbial community of the Inka-Coya sediment core. The core sedimentary radiography, dating, and alpha diversity indices variation along the core depth are displayed.**

The variation in diversity within the sample was measured to determine the changes in the microbial community along the sediment core by calculating the Shannon, Simpson, Chao, and Phylogenetic indices as standard measures of the taxonomic diversity within a sample (Thukral, 2017). The microbial community observed along the core was diverse, based on the DNA samples analyzed. Quantification of diversity showed that it increased with depth, and the Simpson index remained stable after the 2 cm surface layer. At the same time, Chao1 considered low-abundance taxa, and the phylogenetic index based on the



phylogenetic history of the species (Fig. 2). The diversity increases in all cases at two cm long, and maximum values were
observed between 8 and 86 cm, with minor peaks at 96, 120, and 130 cm near the bottom samples.
Three distinct zones in terms of microbial taxonomy could be identified and are statistically different for the four evaluated
diversity indices, except between Zone II and III in the context of phylogenetic diversity (Fig. 3A). These three disjunct
clustering zones of microbial community diversity along the sediment core were also identified in the beta diversity analysis,
where zone I includes the less diverse upper layer (0-8 cm), zone II, the middle zone of the core (9-42 cm) with significant
greater diversity, and zone III the deepest are sampled in this study, depicted in three shades of green going from lighter (zone
I) to darker (zone III) in the PCoA clustering analysis (Fig. 3B).



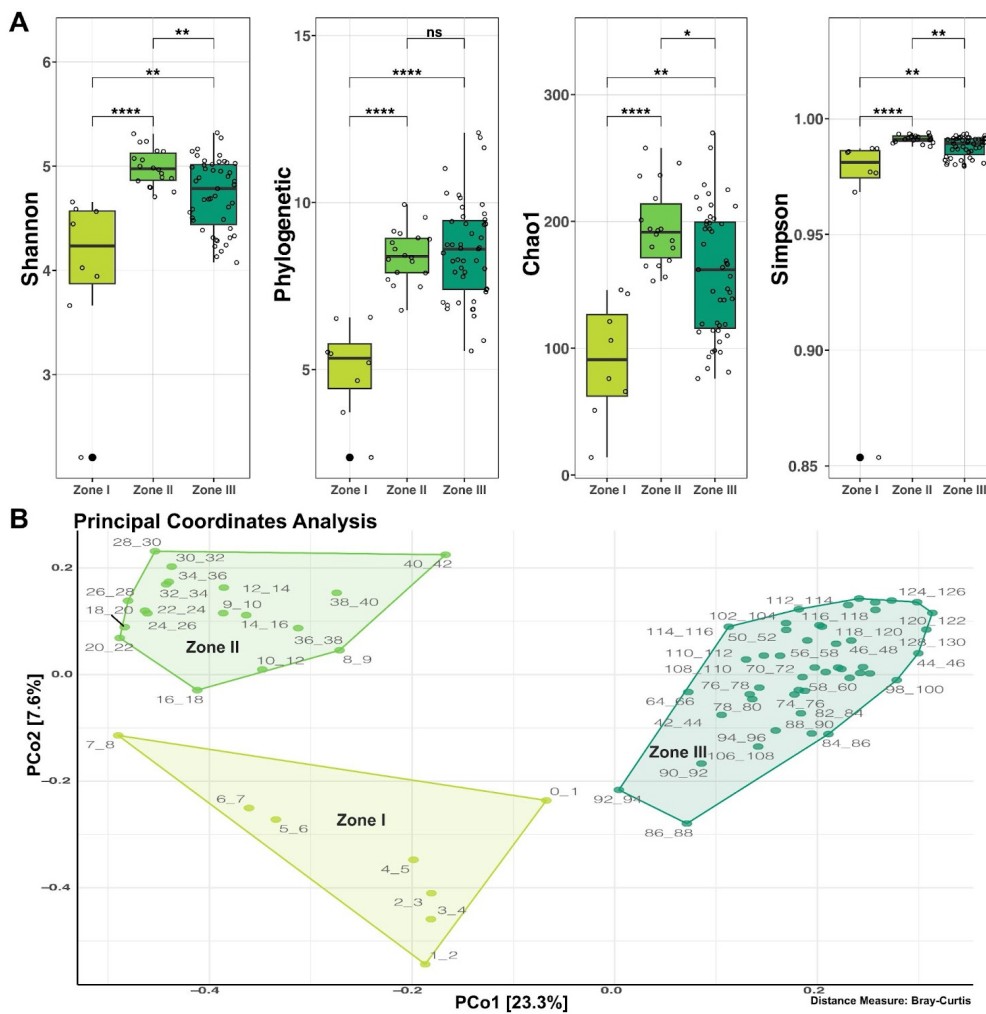

**Figure 3. Clustering of the microbial communities inhabiting the deep sediment of Inka-Coya Lake A) Statistical differences in alpha diversity between depth zones. B) Principal Coordinates Analysis (PCoA) with the ASVs relative abundance using Bray-Curtis as distance metric; each point corresponds to a community, tagged by depth and colored by zone.**

The taxonomic composition and abundance along the communities at phylum rank also reflect the clustering on three distinct zones, where zone I is dominated by Actinobacteriota and includes a great abundance of Firmicutes in the top layers, and Campylobacterota, there is also the presence of Bacteroidota, Halobacteriota, and Plantomycetota. Zone II is more diverse and composed mainly of Campylobacterota, Chloroflexi, Acidobacteriota, and Actinobacteriota. While zone III is the largest and more homogeneous, composed of several low-abundance taxa, dominated by Chloroflexi, Acidobacteriota, and





Actinobacteriota, there is also a higher representation of Crenarchaeota, Nitrospira, Aenigmarchaeota, and Armatimonadota
that in the rest of the zones (Fig. 4).
Notably, 76.6% of the taxa could not be identified at the genus level (0% matched any known species). Thus, Figure 5 shows
the abundance at the "best hit," where Campylobacterota (*Sulfuricurvum*, *Sulfurimonas*), *Mycobacterium*, and *Methanolinea*
dominated the overall community. While in each zone, there are particular taxa associated; for instance, species belonging to
the Aminicenantales Phyla are very common in Zone I; *Pseudarcobacter* is prevalent in Zone II as *Pelolinea* is in Zone III.



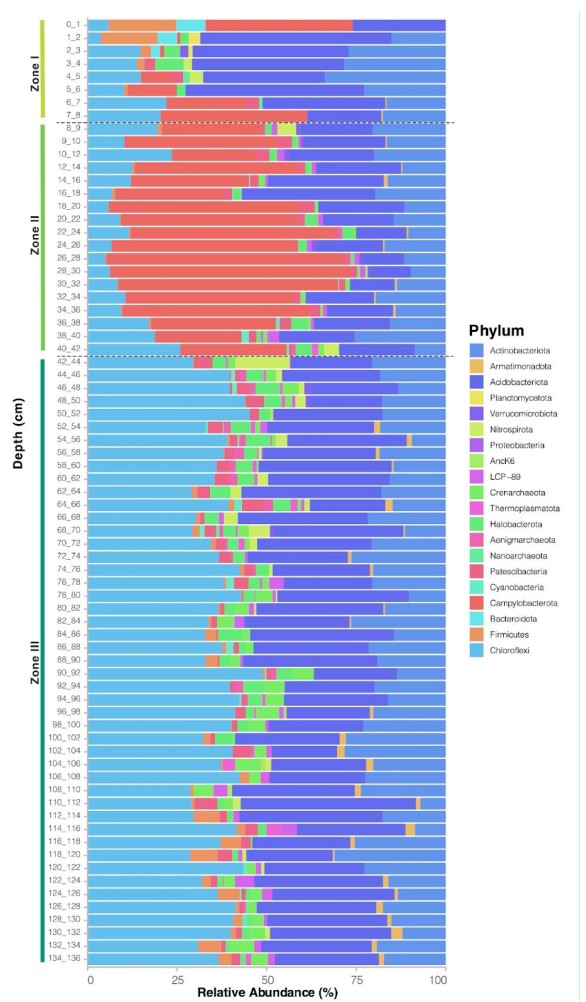

**Figure 4. Taxonomic composition of the microbial community in the deep sediments of Inka-Coya Lake. Stacked bar of the taxonomic composition at the phylum level.**





Figure 5. Heatmap of the abundance of the microbial community at the family level. The color gradient indicates the
abundance of the specific taxa



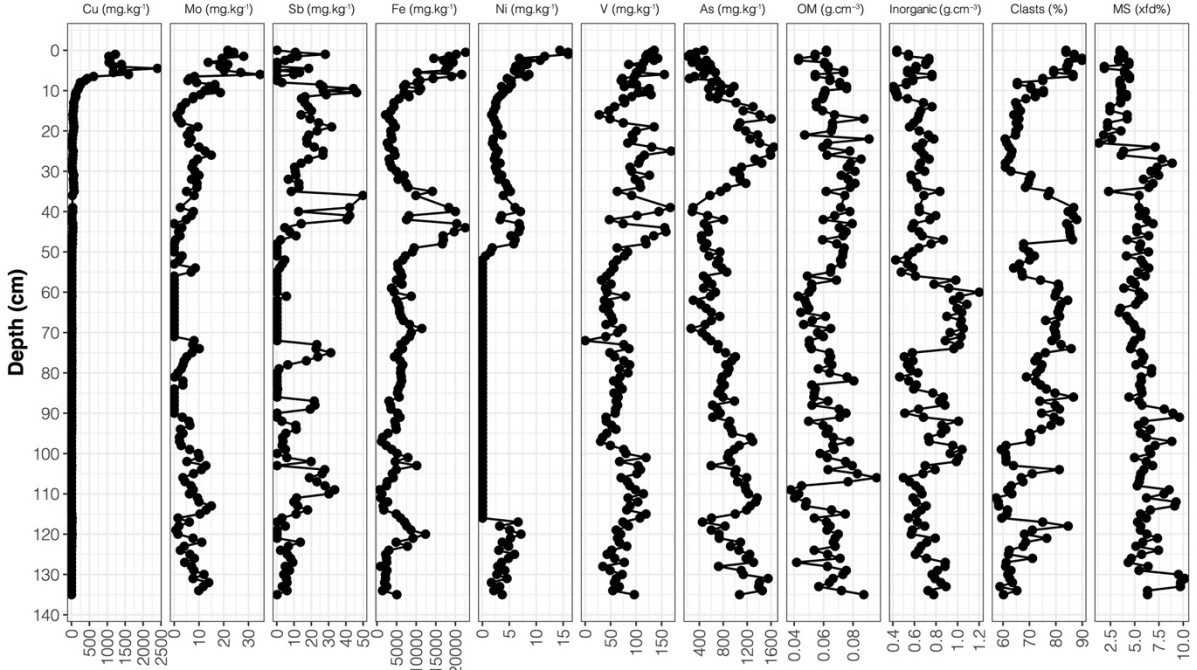

**Figure 6. Depth variation of metals and metalloids concentrations (mg·kg⁻¹), organic matter (OM) and inorganic concentrations (g·cm⁻³), clasts percent, and magnetic susceptibility (MS) measured in Inka-Coya Lake sediment core (modified from Pérez-Portilla et al. 2024).**

Variations in analyzed sediment properties along the core, such as magnetic susceptibility, organic matter, and carbonates/clasts composition, are shown in Figure 6. Copper (Cu) and nickel (Ni) were in lower concentration and variability at greater depths and showed concentration peaks in surface sediments. Elements such as iron (Fe), molybdenum (Mo), and vanadium (V) also showed top sediment peaks. Still, overall, they had more variable behavior than previously mentioned elements in the middle and bottom sediments (> 40 cm). Metalloids, arsenic (As), and antimony (Sb) exhibited the highest concentrations between 10-45 cm depth.

The mean organic matter and carbonate contents are around 8.7% and 19.5%, respectively, while the inorganic density (91%) showed the highest averaged values (0.71 g·cm⁻³). The sediment composition shifts to clay and silt from 28 cm to the top of the core. The content of clasts was predominant in the inorganic fraction, with 71.8% along the core. The carbonate peaks were observed at 10-36 cm, 46-56, and more significant and variable proportions below 96 cm, which did not exceed 40%.

Magnetic susceptibility (MS, χ) values range from -6.09x10⁻⁹ to 8.13x10⁻⁷ m³·kg⁻¹, with an average value of 2.77x10⁻⁷ m³·kg⁻¹. Frequency-dependent susceptibility (χfd%) values range between 1.31 and 10.17%, with an average of 5.62%. Zone I has



the highest values of χ and the lowest values of χfd%, while Zone II shows the lowest χ values and intermediate χfd% values.
Zone III presents intermediate χ values and the highest χfd% values.
Geochemical and magnetic variables are associated with microbial diversity found in the sediment of Inka-Coya Lakes,
differentiated into zones (I, II, and III). Deeper and older fractions of sediments (dark blue), especially Aminicenantae, are
positively influenced by the magnetic susceptibility and inorganic elements in the sediments. Microbial assemblage found in
the middle sediments (Zone II) of the core is driven by organic matter content and water availability, where taxa like
*Sulfurimonas*, *Sulfuricurvum*, and *Dehalococcoidia* were the most represented. The middle zone is associated with a significant
As peak, which suggests that the presence of metal(loid)s positively affects the microorganisms assemblage inhabiting since
middle-to-superficial layers. In Zone II, microbial diversity is mainly associated with low but stable concentrations of organic
matter, a more significant proportion of clasts, and the higher peak of As and Sb in the sediments, where χ values decreased.
Upper Zone I is mainly characterized by metal enrichment, with elevated concentrations of Cu, Zn, Ni, Fe, and Mo, among
other elements. These peaks correlate with high χ values (Fig. 7).

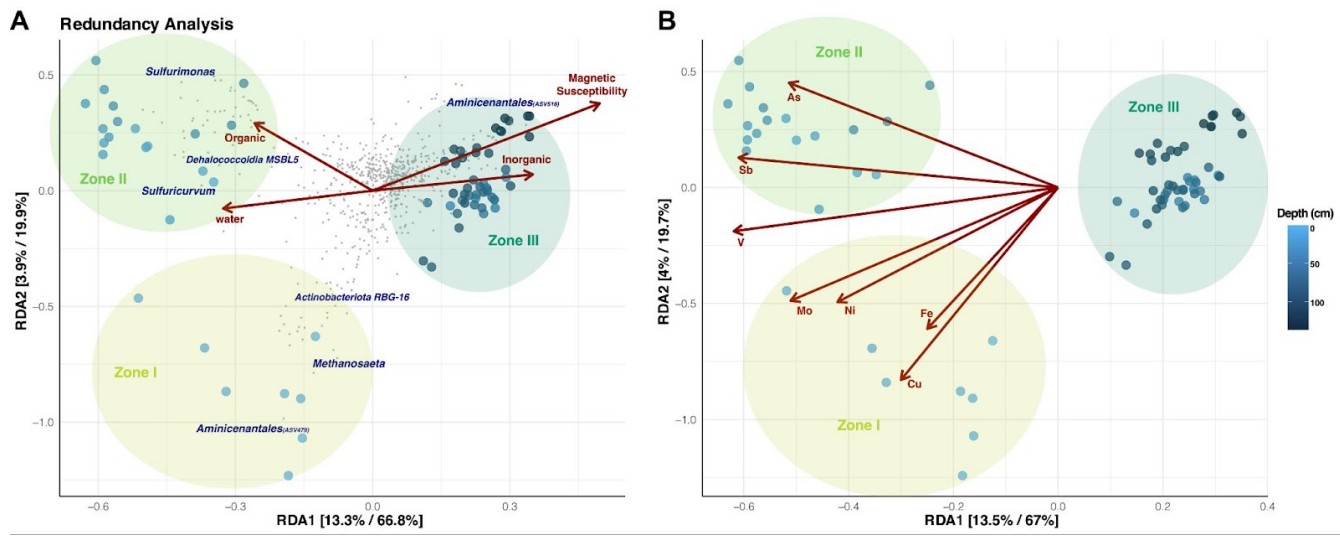


**Figure 7. Redundancy analysis on Hellinger transformed Bray-Curtis distances (corrected by unobserved species) for**
**the microbial communities along the core distance. A) Influence of physicochemical parameters and B) Elemental**
**composition. Depth gradient and Zone parameters were chosen to constrain the multivariate space in a supervised**
**approach. Each axis in the graph shows the percentage of variance explained in an unsupervised and supervised**
**analysis.**
There is statistical significance between some key taxa and the physicochemical and elemental composition along the sediment
gradient in Inca-Coya Lake; for instance, *Methanoregulaceace*, *Ktedonobacteriaceae*, and *Sulfurimonadaceae* are some of the
taxa with the strongest correlation to Cu, Fe, Ni and V presence while zones II and III are the most influenced by these dynamics
(Supplementary Fig. 1).
Regarding metabolic approximation, chemoheterotrophy is the most abundant function in all three zones, while aerobic
chemoheterotrophs are the most prevalent in zone II. Still, chemoheterotrophs (including several electron acceptors) thrive in





zones II and III (Supplementary Fig. 2). Other functions, such as Methanogenesis, were abundant only in deep sediments (zone
III). Moreover, as expected, phototrophy and photoautotrophy were present only in low abundance in zone I, where little light
could reach the community (Supplementary Fig. 2). Methane metabolism is very relevant in all three zones, especially in zone
III, where acetate is the primary source for this pathway (Supplementary Fig. 2). Other relevant forms of energy transformation
are Nitrate reduction VI (assimilation) and starch degradation in zone II. Nitrotoluene degradation and biosynthesis of
unsaturated fatty acids are also crucial in the benthic microbial community as a whole (Supplementary Fig. 2). A somewhat
homogeneous prediction for metabolic ability regarding energy production is found along the communities inhabiting over a
meter deep in the Inka-Coya Lake sediments, where geochemical and magnetic dynamics directly influence microbial
activities.
## 4.  DISCUSSION
Microbial dynamics along the length of the sediment of Inka-Coya Lake are tightly associated with sediment attributes, such
as metal(loid)s concentrations and χfd%, organic compounds- and water- availability. The geochemical characteristics of the
area surrounding Inka-Coya Lake, where active volcanic activity results in the enrichment of elements such as arsenic, sulfur,
copper, and others (Romero et al., 2003; Tapia et al., 2018), suggests that microorganisms assemblage inhabiting the Atacama
area have developed broad tolerance range to this potential toxic compounds. Microbial biomarkers serve as criteria to assess
anthropogenic impact (Yan et al., 2024), and microorganisms can alter the speciation and bioavailability of meta(oids) in an
ecosystem (Niu et al., 2020).
Five stratigraphic zones in the sedimentary core in Inka-Coya were defined from the sediment core here analyzed by Pérez-
Portilla et al. (2024), where the concentration of rock-forming elements such as Cr, Zn, and V are found in concentrations as
expected for the Atacama Desert; while Cu, Mo, Sb and As are higher than expected, suggesting influence from nearby mining
activities (Pérez-Portilla et al., 2024). Copper production in the region generates by-products such as Mo, As, and, to a lesser
extent, Zn (Ramírez et al., 2005; Tapia et al., 2019). Also, mining wastes contain high concentrations of chemical products
such as Pb, Cr, Cd, Cu, Zn, Hg, and Ni, and metalloids, which are often stored in dams or reservoirs (Csavina et al., 2012) or
passed through lotic systems, making them an important source of contamination of inorganic chemical elements for the
aquatic biological communities (Keller et al., 1992; Pollard et al., 2003; Pigati et al., 2011;  Hamilton et al., 2017; Ritter et al.,
2019). Previous studies have shown impacts and metal enrichment from the mining industry in the Antofagasta region
surrounding Calama City and the Loa River basin near the Inka-Coya Lake. Cerda et al. (2019), Vargas-Machuca et al. (2021),
Aránguiz-Acuña et al. (2020), and Zanetta-Colombo et al. (2022, 2024), using both abiotic and biological proxies, have
evidenced an increase in the concentration of metals in different environmental matrices during post-industrial time, attributing
this difference to the mining activities in the area. Additionally, changes in the composition of the zooplankton community
(inferred by diapausing egg banks) and benthic diatoms could be attributed to the increase in Cu concentration evidenced in
the sedimentary cores obtained in this lake.
A strong correlation between mineral composition and microbial diversity in other arid region water bodies, such as salt flats
and brines, is expected (Farías et al., 2014; Castro-Severyn et al., 2021; Dong et al., 2022), as demonstrated in pre-Puna salt
lakes, such as Tebenchique and La Brava (Farías et al., 2014; Ramos-Tapia et al., 2023). The diversity of microbial life in
these shallow salty lakes is dominated by Bacteroidetes, Proteobacteria, and Euryarchaeota (Farías et al., 2014; Fernandez et
al., 2016; Kurth et al., 2021) and hypersaline lakes are mainly composed by Bacteroidetes, Chloroflexi, Cyanobacteria and
Proteobacteria (Dorador et al., 2018).  Inka-Coya, one of the few brackish water lakes located in the Antofagasta Region below





3,000 m a.s.l., and its sediment communities are dominated by Phylum Acidobacteriota, Chloroflexi, Actinobacteriota, and
Campylobacterota, sharing some similarities and taking into account changes in taxonomy (Oren and Garrity, 2021).
At the lower taxonomic rank, the community is dominated by microorganisms with a broad repertoire for mineral interactions,
e.g., there is experimental evidence of organomineralization in extracellular $S^0$ formation by a species of the sulfur-oxidizing
bacteria *Sulfuricurvum* (Cron et al., 2019). Another remarkable microorganism found along the lacustrine sediment is the
cosmopolitan and highly diverse *Sulfurimonas,* which can grow using sulfur, hydrogen, nitrogen, oxygen, and organic
compounds, suggesting it is critical in maintaining trophic dynamics (Han and Perner, 2015). Additionally, *Mycobacterium* is
a saprophytic bacterium commonly found in lakes, rivers, and other water sources (Falkinham et al., 2015), there are some
species representatives of this genus that have bioremediation potential for polycyclic aromatic hydrocarbons (Deng et al.,
2023), suggesting endurance and a broad range of metabolic capacities. There are extensive studies on Mycobacteriaceae
representatives that cause human disease, as reviewed by Falkinham (2009). However, environmental representatives with the
capacity to inhabit poly-extreme environments have yet to be determined; Inka-Coya sediment is an uncharacterized species
of this genus.
Furthermore, several representatives of the candidate phylum Aminicenantales are common in current sediments from Zone I,
i.e., recently deposited sediments during the last ten years. So far, these microorganisms are associated with a fermentative
saccharolytic lifestyle that does not have an isolated representative yet (Kadnikov et al., 2019). While in Zone II, comprising
the period between 10 and 50 years ago approximately, is mainly represented by microorganisms classified as the
*Pseudarcobacter* genus, that was recently separated from the *Arcobacter* genus (Pérez-Cataluña et al., 2018) and are
characterized as mesophilic bacteria that can grow in microaerophilic conditions (Collado et al., 2011). Finally, in the deeper
and older sediments, where anaerobic (or facultative anaerobes) microorganisms can thrive, there is a particular abundance of
an ASV from the *Pelolinea* genus that has only one described species that was isolated from the subseafloor sediment (Imachi
et al., 2014), and an unknown Chloroflexi species that is associated with Dehalococcoidia class a common sub-seafloor
bacterium (Wasmund et al., 2014). This finding suggests past conditions of high salinities for the lake, close to marine
conditions and much higher than the lake's current salinity of around 5 g·L$^{-1}$ (Aránguiz-Acuña et al., 2020). Archaea
representatives found inhabiting the lacustrine sediments include the hydrogenotrophic methanogen *Methanolinea* (Imachi et
al., 2008; Rainey et al., 2015), that are very abundant, especially towards the deeper and more anoxic environment, where
methanogenesis is the central metabolism at play. Overall, there is great taxonomic and metabolic diversity associated with
the microbial community from this lacustrine sediment.
The microbial community in Inka-Coya Lake is primarily heterotrophic with a special enrichment in methanogenic organisms
in the oldest deposits previous to 1950 (zone III), where oxygen levels are lower as evidenced by the metabolic approximation
done in this study, and another kind of metabolism depending of $CO_2$ concentrations, could be dominating. Furthermore, there
is a vast taxonomic novelty harbored in Inka-Coya Lake sediment; over 70% of taxa cannot be identified to the genus level,
indicating there is a significant amount of "microbial dark matter," a term associated with unknown microbial representatives
that can potentially harbor novel bioactive compounds with numerous applications (Zha et al., 2022; Jiao et al., 2021).
The main drivers for microbial community composition in the sinks-Coya sediments were As, Sb, V, Mo, Mi, Fe, and Cu,
which suggest that there are numerous strategies that microorganisms use to resist high concentrations of metal(loid)s that
thrive in this ecosystem (Rahman, 2020; Mathivanan et al., 2021), as observed in the Atacama and Altiplano area (Orellana et
al., 2018; Donati et al., 2019; Aszalós et al., 2020; Castro-Severyn et al., 2019). Additionally, microorganisms can use oxido-
reduction processes to obtain energy from metal ions (Raab and Feldman, 2003; Staicu and Stolz, 2021), and given the known
geochemical characteristics of the area it is expected to find strong relationships between the microbial life and inorganic
compounds, as they can dissolve and precipitate ores and influence metal(loid)s transformations (Raab et al., 2003; Zhou et
al., 2022). In this extreme environment where competition is strong and abiotic pressures are constant, organic matter and



water availability -both parameters critical for most life forms- govern community abundance and composition, suggesting a
delicate dynamic balance reached between abiotic and biotic entities at play. It is important to remark that with the number of
unclassified taxa, a significant number of novel resistant or usage mechanisms remain to be characterized.
Observed trends in magnetic susceptibility in Inka-Coya sediment could be mainly attributed to variations in the concentration
of ferromagnetic minerals, such as titanomagnetite, and authigenic origin minerals, such as sulfide (greigite), as is broadly
explained in Pérez-Portilla et al. (2024) for this sediment core. In this case, the high X values would result from a high
concentration of ferromagnetic minerals of detrital origin, primarily Fe oxides, while elevated xfd% values would be linked to
the presence of greigite of authigenic origin (Pérez-Portilla et al., 2014). Thus, greigite formation typically occurs through the
dissolution of titanomagnetite or other detrital minerals containing Fe (e.g., Chan et al., 2001; Fialová et al., 2006; Versteeg et
al., 1995).
Elevated magnetic susceptibility values in the upper layers of sediments may be attributed to a recent deposition of fine Fe
oxide grains, which could be originating and transported from industrial and urban sources (e.g., Chan et al., 2001); this concurs
with the Fe top sediment peak, and it could be related to a diverse superficial community, while in zone II the lowest values
of X are observed, which could be associated with a mixed community. All these processes occur under a high production of
Fe minerals of authigenic origin (xfd%>3%; Dearing et al., 1996). Moreover, lower levels of Xfd% could be involved in the
production and assimilation of iron sulfides such as greigite (Bazylinski et al., 2001; Lins et al., 2007), promoting a microbial
specialization and increasing resistance of the anaerobic community found in the deeper sediments. Microbial metabolic
responses could satisfactorily support several processes associated with greigite formation. The presence of greigite in the lake
sediments could be associated with reducing or low-oxygen environments (e.g., Benning et al., 2000), where additionally
magnetotactic bacteria would contribute to sedimentary greigite formation through the biomineralization of magnetosomes in
anoxic aqueous environments (Moskowitz et al., 2008).
There are statistically significant correlations between some particular taxa and the geochemical composition along the
sediment gradient in Inca-Coya Lake suggesting that each element directly influences the metabolic capabilities of the
microorganisms and shapes the community selecting taxa that can resist metal(loid)s toxicity (Yao and Gao, 2007; Laplante
et al., 2013; Stankevica et al., 2020; Kostka and Leśniak, 2021; Yan et al., 2020). Further studies that elucidate the functional
properties that these microbial communities have will enhance our understanding in terms of metal(loid)s resistance and the
use of different electron acceptors for energy production.
Finally, increasing metal(loid)s exploitation in the region during the last 100 years has directly influenced their mobility and
the local geochemistry. In this context, microorganisms from these extreme environments are known to be highly adaptive and
have developed several resistance mechanisms and the ability to use these compounds to their benefit. Hence, some bacteria
(and their genes) can be used as biomarkers for the bioavailability of such metals and contamination of soils (Li and Wong,
2010; Roosa et al., 2014). In Inka-Coya Lake, a selective process could be evidenced along the sediment record, in which the
recent period is characterized by a marked increase in chemical elements and microbial composition, which could be associated
with the increasing mining activity and other anthropological activities, as water extraction or aridity increase by climatic
changes, that also would increase the disturbance of this relevant area in the core of the most arid non-polar Desert.





# 5. CONCLUSIONS

This study represents the first to encompass a deep gradient of microbial life in a desert lake in the Atacama area, proposing a biological clustering of taxa and function in three periods that stratified for over 600 years, since the pre-mining period, the mining development and the most industrialized mega mining observed nowadays. A great taxonomic novelty exists among the microbial community inhabiting lacustrine sediments of Inka-Coya that potentially holds an abundant novel repertoire of bioactive compounds of biotechnological interest. Mineralogical enrichment, water, organic matter availability, and magnetic susceptibility are also variables that explain the changes within the microbial community. There are strong relationships between geochemical composition and microbial diversity, especially in Cu, Fe, Ni, and V. The first zone is less diverse and dominated by Actinobacteria; the second zone has a high abundance of Chloroflexi, Acidobacteriota, and Actinobacterota. The third zone shows more rare taxa with lower abundance and clusters the higher area of the studied sediments, including archaea. Overall, chemoheterotrophy is the prevalent energy production mechanism along the sediment core. This unique and fragile ecosystem depends on biogeochemical dynamics vulnerable to anthropogenic activities and climate change.

**Appendix A.**

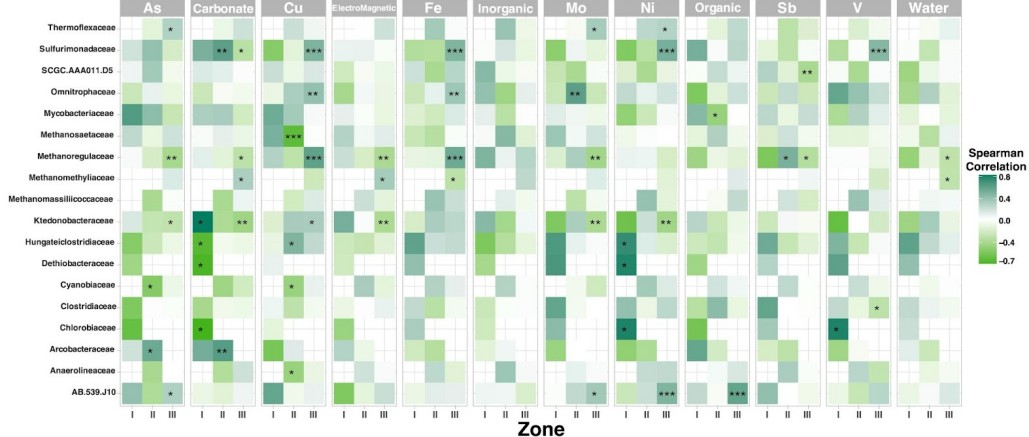

**Figure A1**. Pearson correlation between the top microbial families and geochemical parameters. Asterisks show the level of significance (\*$p$ value < 0.05, \*\*$p$ value < 0.01, Pearson correlation). Purple, blue, and white indicate positive, negative, and no correlation, respectively.

**Appendix B**





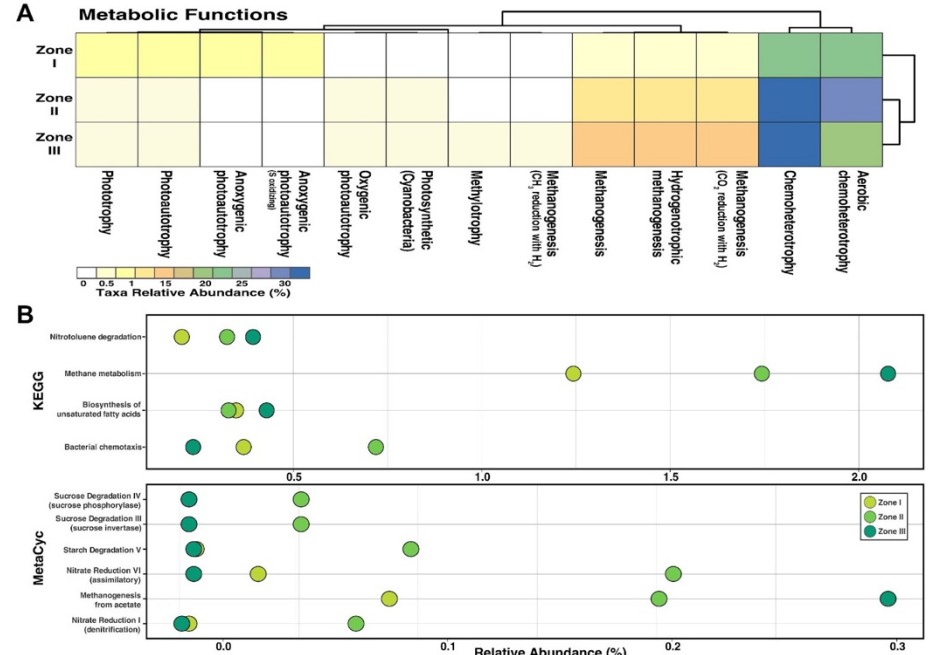

**Figure B1**. Metabolic pathways prediction of the microbial community inhabiting sediments along a depth gradient in Inka-Coya Lake. The predictions are based on the identified taxonomic composition according to the comparison with different databases, such as: A) FAPROTAX; B) KEGG and C) MetaCyC. Categories with significant differences (p<0.05) according to Kruskal-Wallis test are displayed.

**DATA AVAILABILITY STATEMENT**

The raw sequencing data presented in this study have been deposited in the DDBJ/ENA/GenBank SRA database under the BioProject: PRJNA1067596.

**AUTHOR CONTRIBUTIONS**

Conceptualization: AAA; FR, JCS, CPE. Data curation: CPE, JCS, FR, IHF, AAA. Formal analysis: CPE, JCS, FR, AAA, AM, HP. Funding acquisition: FR, AAA Methodology: CPE, JCS, AM, HP, FR, AAA. Supervision: AAA, FR Writing original draft: CPE, JCS. Writing review and editing: AAA, AM, HP, JCS, CPE. All authors have read and agreed to the published version of the manuscript.

**FUNDING**

ANID 2020 FONDECYT Regular 1200423 (AA-A), UTA-Mayor 9738-24 (AA-A)

ANID 2022 FONDECYT Regular 1220902 (FR)

ANID 2023 FONDECYT postdoctoral 3230189 (CPE)



ANID 2021 FONDECYT postdoctoral 3210156 (JCS)

**COMPETING INTERESTS**
The contact author has declared that none of the authors has any competing interests.

**ACKNOWLEDGMENTS**
Likan antay Community of San Francisco de Chiu-Chiu.
MAINI-UCN (Kappabridge)

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
