# Peer review of "Microbial communities inhabiting 600-year-old sediments in the Inka-Coya Lake located in the"

_EGUsphere, 2024_

## Author Response (AR1)

We thank the reviewers for their time and suggestions that have improved our manuscript. Here we present a point by point response to each raised concern.

Reviewer 1

Pardo-Esté *et al*. provide a biogeochemical and microbiological survey of a single sediment core (LIC-SHC03) from Inka-Coya lake, located in the Atacama Desert. The authors provide a time model that, they propose, resolves a local anthropogenic impact sequence generated by copper mining. The authors highlight that a large amount of the microbial community, inferred by 16S rRNA gene sequences, belong to unclassified organisms and proffer statistical correlations between certain elements and sequence representatives related to chemoheterotrophic lineages. Overall, the study is due to the site and context is novel and provides a valuable contribution to environmental impact assessments associated with copper mining broadly and anthropogenic effects on relatively isolated lake ecosystems specifically.

General Comments/Suggestions

The writing contains numerous grammatical errors and may benefit from additional revisions by an additional number of native English speakers. I've done my best to highlight some of those instances, but many such issues remain. The study's strength would largely benefit from additional details about the precautions the authors undertook during the molecular analysis step (details below). The taxonomy-based functional prediction approach (PICRUSt2), as noted in the methods, is tenuous at best. Thus, the tone of certain parts of the discussion needs to reflect this potential interpretative vulnerability.

R: We thank the reviewer for the suggestions, which have improved the manuscript. A new grammatical review was done by a certified English proofreading service and the whole manuscript was adjusted. Also, we included more details in the methodology and the tone of the metabolic conclusions have been adjusted.

Specific Comments/Suggestions/Questions

L34-37: A bit of a run-on sentence, please revise for clarity.

R: We have improved these sentences in the revised manuscript.

L58: What do the authors mean by "perform"; perhaps live or thrive may be better choices here?

We have changed this word for "thrive" to indicate that microorganisms not only survive but are able to efficiently use the resources available to develop.

L82: Do the authors mean assembly instead of assemble?

We have changed the wording to community.

L82-85, L117-122: Are there other well established sedimentation rate models in other lakes of the region? Do you they match your independently determined rates?

R: In the Atacama Desert, there are no water bodies forming lakes, properly speaking. The Lakes existing at the Inka Coya latitude, such as Miscanti or Miñiques, are located at an altitude of >4000 m above sea level, in the Andean Altiplano (23°43′30″ S, 67°45′54″). Because of the wide climatic differences between the Altiplano (humid in summer) and the aridity of the desert and pre-puna, the models developed for these systems (Grosjean et al. 2001), are not comparable with those described for the Chiu-Chiu locality. In this sense, the development of a model in the Inka Coya Lake, with a depth of 18 m, suggests even better conditions for preserving the sedimentary records than what can be achieved in lagoons such as Miscanti, which is much shallower (8 m).

Grosjean M, van Leeuwen J.F.N., van der Knaap W.O., Geyh M.A. et al. A 22,000 14C year BP sediment and pollen record of climate change from Laguna Miscanti (23ºS), northern Chile. Global and Planetary Change 28: 35–51.

L144: How exactly were the 5g of sediment collected for molecular analysis? Where these frozen split cores? Mini-cores with syringe? Specific precautions for prevent cross-contamination (top to bottom sequence) or external contaminants (subsampled in laminar flow hood)? Please expand on the specific extend of aseptic technique used.

R: This was clarified in the corresponding methods section. See revised version: For sub-sampling, the frozen core was sliced into sections every 0.5 cm to a depth of 12 cm and every 1 cm thereafter, resulting in 146 sediment samples. For molecular analysis, sediment sub-samples were taken from the center of each segment under a laminar flow hood using ethanol-sterilized tools. These sub-samples were stored in sterile 15 mL tubes, labeled, and kept frozen. Additional sub-samples were collected for geochemical and magnetic susceptibility analyses.

L145: How was the DNA integrity, quantity, and quality ultimately determined? (ABS ratios?, smear gel?, what were the concentrations found for your samples?).

R: DNA integrity was addressed by running the samples on an 1% agarose gel to see clear bands with no smear. Quantity was determined using the Qubit4 fluorometer with the Qubit 1X dsDNA HS assay kit, where concentrations ranged from 1.18 ng/ul to 47.90 ng/ul. Also, contamination was measured using the 260/280 nm ratio.

L149: was there a specific reason to not use the new version of the Earth Microbiome Primers (Parada et al.) given that the revised primer may be better able to capture Archaea?

R: The used primers are the standard offered by the sequencing service, which correspond to those described in Herlemann et al., 2011.

Bakt_341F (CCTACGGGNGGCWGCAG)

Bakt_805R (GACTACHVGGGTATCTAATCC).

Herlemann, D. P., Labrenz, M., Jürgens, K., Bertilsson, S., Waniek, J. J., & Andersson, A. F. (2011). Transitions in bacterial communities along the 2000 km salinity gradient of the Baltic Sea. The ISME journal, 5(10), 1571-1579.

L151: Where there any blanks sequences along with this analysis. If so, how were those sequences treated?

R: We used blank control for every batch of DNA extraction, in every case the quantification by Qubit was "too low" and no bands were observed in agarose gels. This processing had a low contamination risk, as it was carried out under sterility conditions and inside a laminar flow hood. Also, the samples for molecular biology were the first to be taken and selected from the most inner part of the core (which had not been exposed or touched by another instrument).

L159: Is this relative abundance normalized in any way before rank visualization in ggplot2? (Rarefied?). Please explain.

R: Yes, we use the Variance Stabilizing Transformation to account for the differential sample size (reads number) and data inner variability. The rarefaction method was not used. This information has been added to the revised manuscript.

L163: Can the authors clarify the "phylogenetic diversity" metric?

R: Phylogenetic diversity is equal to the sum of the phylogenetic tree branch lengths, as they represent the relative number of new features arising along that part of the tree.

Faith DP (1992) Conservation evaluation and phylogenetic diversity. Biol Conserv 61:1–10

L164: Did the authors perform any multiple testing correction following the non-parametric Wilcoxon test (e.g.: Pval adjustments Benjamini-Hochberg, etc.)?

R: Yes, Bonferroni correction was applied, this information has been added to the revised version: Wilcoxon statistical tests with Bonferroni correction were performed to compare means between the identified zones, with results visualized using the ***DESeq2*** v1.42.0 (Love et al., 2014) and ***ggpubr*** v0.6.0 (Kassambara, 2017) packages

L169: Any post-hoc multiple comparison corrections on your ANOVA test involving geochemical gradients to address potential type-1 error inflations (e.g.: Bonferroni, Tukey's HSD?).

R: In this case no Post-hoc correction was applied, as we performed this analysis using the default and recommended parameters, considering that the vegan package has been extensively validated and benchmarked.

L174: Again, I think PICRUSt is a great hypothesis-generating approach, particularly for synoptic predictions of potential metabolisms within a single sample; however, statistical approaches described here, aiming to infer comparative differences across samples of taxonomically-inferred pathways and statistical differences between the estimates of these inferred pathways is tenuous operation. Perhaps this part of the methods would be a great part to explain this to the readership? I believe that would help the strength of the paper. Note: I see that the authors encourage further work with metagenomic methods in the discussion as well.

R: Indeed, PICRUST performs an inference that can give us clues about what may be happening in our communities at the level of functional potential and precisely we used it as an approximation that needs to be confirmed. Nonetheless, this information used carefully could guide our research and also help us to raise new questions and experimental designs, such as using shotgun metagenomics. This methods section was rephrased to improve the readers understanding, as we wanted to identify functional potential differences among the three identified core zones. See revised version: Differential abundance of inferred pathways across the three identified zones along the core, which presented significant differences in taxonomic composition, was tested using the Kruskal-Wallis test (confidence interval = 0.95) and the Benjamini-Hochberg false-discovery rate correction using the ***ggpicrust2*** v1.7.2 R package (Yang et al., 2023).

R: L277-279: Do the authors mean "communities living in depths greater than a meter"?

We have improved clarity in this sentence in the revised manuscript. Revised manuscript: among microorganisms inhabiting sediments deeper **than** one **meter** in Inka-Coya Lake

L284: change "microorganisms assemblage" to "microbial assemblages".

R: Thank you for pointing this out, we have changed it.

L311: Can you specify the "lower taxonomic rank".

R: We have included some examples at the genus level. Revised manuscript: Phylum Acidobacteriota, Chloroflexi (*Pelolinea*, among others), Actinobacteriota (*Mycobacterium*), and Campylobacterota (*Sulfuricurvum* and *Sulfurimonas*)

L318: Do the authors mean viability rather than endurance?

R: We mean adaptability, in the sense of metabolic flexibility and the ability to use different compounds as a source or energy. We have changed the word.

L320: This may be an uncharacterized species of the genus since this based on a 16S rRNA gene study rather than culture, isolation, and characterization as would be necessary to make the statement as is. Please reframe to address this.

R: This sentence was deleted.

L332: Does your figure 4 show a high abundance of this methanogenic lineage as these other studies? It looks like, if you got it in your cores, it is much less prevalent correct?

R: Figure 4 shows the distribution of the taxonomic composition at the phylum level. We have also included more detailed information on methanogenic lineage in the revised manuscript. Revised manuscript: …which was distributed along the length of the core, with abundances ranging from 0.7% to 2.3%. This archaeon is particularly abundant in older, more anoxic environments (Zone III, dating from 1400 to 1950), where methanogenesis is the primary predicted metabolic pathway

L337: Perhaps chance "special" to "predicted" here.

R: We added "predicted" metabolisms.

L338: A metabolic approximation is not physiological evidence. Please re-write this to address this.

R: We specified that in this study we describe the "potential" metabolic diversity as it is an approximation based on taxonomy.

L340: Please note that in many other subsurface environments (terrestrial, marine from my experience) as deeper layers are sample lower percentages of 16S rRNA gene lineages are able to be identified (have close relatives) at the genus level. Thus, this observation falls in line with similar underexplored environments (Orca Basin, for example). This means that the deep community of this lake may not have an increased amount of "microbial dark matter" relative to any other under explored environment (most of the habitable space on Earth). I'd encourage to authors to re-contextualize their observation mentioned here accordingly.

R: We agree with the statement of the reviewer, more extreme and less studied environments have greater taxonomic novelty and there are many ecosystems on Earth that have such characteristics and harbor microbial dark matter. However it is a fact that in this study that with the current scientific knowledge and databases, the microorganisms inhabiting deeper sediment in Inka Coya, that had not been sampled before, constitute many unknown species and this has a great impact on efforts to characterize and protect the local natural patrimony.

L369: Did the authors observe any classic magneto-tactic lineages in their data?

R: We changed "would" to "could" as there are not described magnetotactic bacteria, however taking into account that this group is understudied and many do not have a taxonomic characterization and the magnetic composition found in Inka Coya is is possible that novel or uncharacterized species appear with culture-dependent efforts to isolate magnetotactic bacteria.

L375: This reads a bit ambiguous. Can the authors explicitly state the need for shot-gun metagenomics.

R: We have stated this in the revised manuscript.

L380: What do the authors suggest by resistance (a term often misconstrued in microbiology to mean antimicrobial resistance): perhaps detoxification and/or adaptations? To what environmental insult specifically?

R: We have improved the sentence to better explain the point of the need to understand the mechanisms at play and the functional potential held by these communities. Rv: Further studies that elucidate the functional properties of these communities using shotgun metagenomics could enhance our

understanding of the mechanisms used to resist and survive environments with high metal(loid) concentrations and the ability to employ different electron acceptors for energy production.

L389: Change function to "predicted function".

R: This was changed in the revised manuscript.

L396: No sure what authors mean by the "higher area" of studied sediments? Please disambiguate as surficial/interface, mid-core, and bottom core consistently throughout text to avoid reader confusion.

R: We have specified that we mean the sediments closer to the surface.

Reviewer 2.

The authors studied the microbial community compositions and their response to the environment in sediments of Inka Coya Lake located in the Atacama Desert. They found that the microbial community in lake sediments contained over 70% of unclassified organisms, highlighting the novelty of microbial taxonomy in this ecosystem; Microbial communities assemble in three different regions: surface communities, intermediate and mixed communities, and anaerobic communities found exclusively in deep sediments. The microbial composition consists mainly of chemically heterotrophic bacteria, which are closely linked to methane metabolism; There is a strong correlation between certain subgroups (such as Sulfurimonadaceae, Metanoregulaceae, and Kdonobacteroceae) and elements such as Cu, As, Fe, Ni, and V, as well as the magnetic properties of the environment. These results are worth publishing. However, the authors must revise their manuscript thoroughly in the following aspects before it can be accepted:

Abstract

Line16-20: Authors should briefly explain the scientific problem or scientific knowledge gap that this article aims to address, rather than highlighting the characteristics of Inka Coya Lake here.

R: We thank the reviewer for the suggestion, we have edited the abstract and the whole manuscript with the assistance of a professional english proofreading service. Rv: We aimed to study the microbial community dynamics in Inka-Coya Lake, located in the Atacama Desert, where active geological activity and the local mining industry influence biological dynamics in this ecosystem, as suggested for macroinvertebrates and plankton communities in the lake.

Line 28-30: Authors should tell readers what significance and/implications of their results are, rather than what to do in the future.

R: Following the suggestion, we have reinforced the main findings of this investigation. Rv: These results highlight the strong correlation between geochemistry and microbial life, which could be disrupted by continuous mining activity in the area.

Due to the word limit of the abstract, it is not appropriate to use a large amount of text here to introduce the background information on the lake. A standard abstract should include the following parts: the scientific problem or scientific knowledge gap to be solved, the sample (research object) of the study, the research methods, the main research results, and the scientific significance of these research results. However, there should be no inferential sentences in the abstract. Therefore, the authors need to rewrite this abstract.

R: We have shortened the description of the site in the abstract.

Introduction:

The author should provide the necessary scientific hypotheses or the scientific problem you are solving, which is currently missing. This does not conform to the logic of a scientific paper.

R: We have stated the hypothesis of this investigation in the final paragraph of the introduction. Rv: This study aimed to characterize the microbial community along a lacustrine sediment core, capturing the depositional history of the Inka-Coya Lake over the last 600 years. We hypothesize that the microbial community assemblage of Inka-Coya Lake is strongly associated with sediment attributes shaped by autochthonous and allochthonous processes, particularly anthropogenic contributions from the metal-mining industry that has operated **near the lake** for the past century

Results:

I didn't understand why the author provided data on the age of sediment core. The absence of core age data does not affect readers' understanding of the composition of the microbial community in the sediment core and its response to environmental factors. My understanding is that sediment core age data should be linked to human activities and the resulting environmental changes. However, this part of the data is not presented in the manuscript. Otherwise, the sediment core age testing method and associated result data can be deleted.

R: The sediment core data is relevant for the study, first regarding the local environment and the changes that the area faces with initiation of great-scale mining industry and human development overall, as this isolate and fragile lake is exposed to changes and the local microbial life is highly adapted to survive in particular extreme conditions thus subtle changes would strongly affect the community dynamics. Therefore in our investigation we

aimed to determine how sedimentation associated with metal(oids) deposition over long periods of time would influence modern microorganisms inhabiting the lake sediment.

Line 192: Figure 2, If authors compare different indices of diversity, it is necessary to explain in detail the mathematical and ecological differences between the indices, because they appear to express the same meanings.

R: Microbial alpha diversity can be measured with different indexes and the sum of arguments allows to reach a conclusion. We reference the following review with detailed information on the specific of each index. Thukral, A. K. A review on measurement of Alpha diversity in biology. Agri Res J, 54(1), 2017

Line 232: Figure 6, authors should put a time scale on this graph instead of just showing the depth coordinates. In addition, if the author wants to discuss the response of the microbiome to heavy metal (such as the first part of the discussion section), it is best to show the picture of the metal element and the microbiome under the same depth/time scale, and mark the location of important nodes (such as human activities, climate events, etc.) for easy understanding.

 R: We have included a time scale on Figure 6.

Discussion:

In the discussion, the author should provide answers to the scientific hypotheses or questions raised in the previous introduction, thus ensuring coherence between the two. Currently in the discussion there is no author's viewpoint or conclusion, but only a repetition of the previous results.

R: We appreciate the suggestion. We have better organized the discussion and have incorporated sentences that allow us to strengthen the impelling findings that give more coherence to the text, while at the same time, they are better associated with the objectives and hypotheses stated in the introduction. We believe that the text gains in integrity with these changes.

Line 343: It is suggested that the authors use statistical methods to further quantify the effects of different elements on the composition of microbial communities and discuss the differences between different depth regions. Because As, Sb, V, Mo, Mi, Fe and Cu have great differences in physical and chemical properties, biological activity and biological toxicity, it is not appropriate to generalize.

R: Supplementary Fig1A shows the taxa statistically correlated with each element separately, as calculated by Pearson correlation. We acknowledge that each element has particular physicochemical properties, as well as bioavailability and activity and their interactions per se can be investigated in future works that include metagenomics, MAGs and cultivated

representatives from these communities, however this falls outside the scope of this investigation.

Supplementary:

Line 408: The subgraph showing MetaCyC is missing the mark "C"

R: This is part of the B panel, we have corrected the figure caption.

---

## Author Response (AR2)

We would like to thank the editor and reviewers for their time and comments that have ultimately improved our manuscript.

Next, we list a point-by-point response to the concerns raised by Reviewer 2

1. Introduction:

    1) The introduction provides a good overview of the Atacama Desert and its unique environmental conditions. However, it could benefit from a more detailed discussion of previous studies on microbial communities in similar desert lakes, highlighting the gaps that this study aims to fill.

    We are thankful for the reviewer's suggestion. We have included references for studies carried out in other desert lakes, including a review that reflects previous investigations done in different locations worldwide. However, there is an important gap in knowledge regarding the unique Atacama Desert lakes, given their local geochemistry and the intense mining industry maintained in the area; the few studies from the Atacama Desert lakes were already included in the previous version of this manuscript. Furthermore, we agree with the reviewer that a detailed discussion would enrich this study, and more perspective has been added, for example, highlighting the methanogens role that is in accordance with our results. See lines 87-92 and some discussion sections in the revised manuscript.

    2) The introduction could also include a clearer hypothesis or research question, guiding the reader through the study's objectives.

    We have now added lines 101-102 to emphasize that our research question aimed to describe geo-microbial dynamics and establish a baseline study in this underexplored lake under constant anthropological pressures.

2. Methods:

    1) The methods section is detailed and well-structured, providing clear descriptions of the sampling, DNA extraction, and sequencing procedures. However, the section could include more information on the quality control measures used during sequencing and data analysis to ensure reliability.

    We have included more details of quality control in lines 157-158, also the section: "Sediment sample processing and DNA extraction" details precautions and measures taken to ensure the samples were not cross-contaminated. Also, during the sequencing data processing, strict quality controls were carried out on the raw data (line 167) and the resulting analysis to avoid overestimations and misinterpretations (170-171). We also used validated methods, pipelines and packages to ensure reliability and replicability of our results.

3. Results:

    1) The results are presented clearly with detailed figures and tables. The identification

of three distinct microbial zones is a significant finding, but the study could benefit from more detailed statistical analyses to support these observations.

We acknowledge the reviewer's comments. From the physical-chemical properties point of view, our group statistically determined these zones in a previously published paper (Pérez-Portilla et al., 2024) using a data integration model (which we refer to in figure 6). In this work, the taxonomic composition analysis reflected the zone differentiation (which is supported by Wilcoxon statistical tests to compare means between the identified zones for all tested Alpha diversity indices). Additionally, beta diversity analysis using Hellinger transformed Bray Curtis distances shows a clear segregation of the samples into well-defined groups, confirming also all previous findings. On the other hand, the significance of the selected geochemical variables was also tested statistically with ANOVA. This is detailed in lines 177-185 and can be observed in Figure 3.

2) The correlation between microbial diversity and geochemical properties is compelling, but the study could explore potential mechanisms underlying these correlations, such as specific metabolic pathways or ecological interactions.

We initially did not explore these underlying mechanisms further since the microbial community's functional potential is out of the scope of this study. Our results focus on taxonomic composition and abundance; additionally, most of the microorganisms found along the sampled core are unknown or yet to be classified; this novelty prevents partial conclusions based on lower taxonomic groups. However, in the results and discussion sections, we have included more details based on previous investigations and our metabolic estimation based on the taxonomic composition. See lines 231-237, 240-241

4. Discussion:

1) The discussion effectively links the findings to broader ecological and environmental contexts. However, it could delve deeper into the implications of the study for understanding microbial adaptations to extreme environments and potential applications in bioremediation or astrobiology.

We have included statements that suggest ecological interactions among the microbial communities and their importance in the study of the origin of life, life in other planets, and potential biotechnological importance of these microorganisms that are very under-characterized. See lines 334-339

2) The discussion could also address potential limitations of the study, such as the reliance on a single sediment core and the assumptions made in functional predictions.

We have also highlighted the limitations of this study in the revised manuscript. See lines 371-376.